



# FarmCan: A Physical, Statistical, and Machine Learning Model to Forecast Crop Water Deficit at Farm Scales

Sara Sadri[1], James S. Famiglietti[1], Ming Pan[2], Hylke E. Beck[3], Aaron Berg[4], and Eric F. Wood[†]

[1]University of Saskatchewan, Global Institute for Water Security, SK S7N 3H5, Canada
[2]Scripps Institution of Oceanography, UCSD, La Jolla, CA 92093, U.S.A.
[3]Joint Research Centre of the European Commission, Ispra 21027, Italy
[4]University of Guelph, 50 Stone Road East, Guelph, ON N1G 2W1, Canada
[†]deceased, Nov 2021

**Correspondence:** Sara Sadri (sara.sadri@usask.ca)

**Abstract.**

   In the coming decades, a changing climate, growing global population, and rising food prices will have significant yet uncertain impacts on both water and food security. The loss of high-quality land, the slowing in annual yield of major cereals, and increasing fertilizer use, all indicate that strategies are needed for mon-

itoring and predicting ongoing and future water deficits on farms for better agricultural water management decisions. Most such activities are based on in-situ measurements which are costly, hard to scale, and ignore the wealth of spatial and temporal information from remotely-sensed data. In this study, we designed Farm-Can, a novel and robust climate-informed machine learning (ML) framework to predict crop water demand at the farm scale with up to 14 days lead time. We use a diverse set of simulated and observed near-real-time

(NRT) remote sensing data coupled with inputs from farmers, a Random Forest (RF) algorithm, and precipitation (P) prediction from MSWEP to predict the amount and timing of evapotranspiration (ET), potential ET (PET), soil moisture (SM), and root zone soil moisture (RZSM). Our study shows that SM and RZSM are the variables that are more correlated with P, while PET and ET do not show a strong correlation with P, SM, and RZSM. Our case study of 4 farms in the Canadian Prairies Ecozone (CPE) using $R^2$, RMSE, and

KGE indicators, shows that our algorithm was able to forecast crop water requirements 14 days in advance reasonably well. We also found that during 2020, RF forecasted ET and PET and needed irrigation (NI) with more accuracy than SM and RZSM, although this might vary based on the soil type, location, year of study,





and crop type. Due to the forecasting capability and transferability of the mechanism developed, FarmCan is
a promising tool for use in any region of the world to help stakeholders make decisions during prolonged peri-
ods of drought or waterlogged conditions, schedule cropping and fertilization, and address local government'
policy concerns.

## 1   Introduction

The FAO estimates that global food production must increase 50-70% by 2050 to feed the growing popula-
tion of 9.1 billion (UN/ISDR, 2007; FAO, 2009). Combined with the increasing frequency of drought due
to climate change, non-sustainable use of groundwater, and increasing competition from municipal, environ-
mental, and industrial water needs, farmers are facing the challenge of maximizing crop production without a
growing water supply (Han et al., 2018). Farmers, however, may lack adequate means and incentive to char-
acterize crop water use, and thus agricultural water management often operates under conditions of unknown
water deficiency (Levidowa et al., 2014). Needed Irrigation (NI) is the amount of water to satisfy crop water
demand and is a critical measure for enhancing agricultural Water Use Efficiency (WUE) (Kirda, 2000). In
irrigated farms, information on NI can help in regulating water deficit to achieve higher levels of crop pro-
duced per unit water consumed (Han et al., 2018; Chalmers et al., 1981). However, information on NI is also
important for rainfed farms as it gives farmers incentives for more efficient practice, and helps them adapt to
climate change by implementing viable solutions to gain more benefits from farming(Levidowa et al., 2014;
White et al., 2020). Therefore, the development of tools that enable estimation of NI is critical for farm deficit
water management strategies to minimize potential crop failure and losses (White et al., 2020; Levidowa et al.,
2014; Th.F.Stocker et al., 2013; Geerts and Raes, 2009).

Although hydrological practices have significantly advanced the study of large catchments for water re-
sources purposes, they have—to date—limited implications for real-time agriculture and farm-scale NI (Jia
et al., 2011). The major portion of agricultural studies has focused on model-based crop water stress, mostly
because of the difficulty associated with measuring water availability for specific agricultural periods such as
crop growth or yield (Ash et al., 1992; Wittrock and Ripley, 1999; Quiring, 2004). Some studies (Smilovic
et al., 2016; Andarzian et al., 2011) employed the crop-water model, Aquacrop, to evaluate the timing and
spatial distribution of irrigation water between farms within a watershed in western Canada. They showed that





wheat production alone could be maintained while reducing water use by 77% and production could increase

by 27% without increasing irrigation water use. Despite their advantages, crop water stress models can be too

complicated to operate and cannot easily be used as operational forecasting tools due to the limited spatial

and temporal availability of models' input data. Plant hydraulic models, for example, have relatively complete

mechanistic representations of humidity, temperature, and Leaf Area Index (LAI), but they are usually too

complex, with many parameters that are hard to measure for crops (Yang et al., 2020).

In recent years, Machine Learning (ML) models have become a useful calculation tool for minimizing

farm crop loss. ML models can learn from training data and construct regression and classification models

for multivariate and non-linear systems. Combined with Near-Real-Time (NRT) remote sensing, farm-level

information with reasonable confidence and with potential for better-informed water resources management

is now achievable. Remotely-sensed data are especially useful for areas where more advanced, on-farm tech-

nologies may be too costly. The use of ML for the potential evaluation of water stress continues to be under-

explored and the existing methods can still be greatly improved (Virnodkar et al., 2020; Yang et al., 2020;

ScienceDaily, 2021).

In this study, we focus on combining NRT remote sensing observations from various platforms and farm-

specific information with ML and stochastic analysis to develop FarmCan. FarmCan is a hybrid physical-

statistical-ML model for agricultural applications trained on surface soil moisture (SM), root zone soil mois-

ture (RZSM), precipitation (P), Evapotranspiration (ET), and Potential ET (PET) to monitor and forecast NI

on daily basis and up to 14 days in advance. Our goal is to develop the FarmCan model to (i) use farm coor-

dinates, crop type, and the length of the growing season as user inputs, and retrieve and analyze NRT remote

sensing data at the farm scale in real-time; (ii) establish a methodology to forecast PET, SM, and RZSM using

P prediction; and (iii) develop a climate-informed forecast of subfield crop NI volume and its timing with up

to 14 days lead time. Our analysis and framework are developed for the Canadian Prairies Ecozone (CPE)

farms but can be transferred anywhere to inform farmers and other stakeholders where and when additional

water is potentially needed to compensate for water deficits. FarmCan also provides information that can be

useful to governments, water managers, agriculturalists, and industries' sustainable initiatives to grow more

food with better-managed water. The remainder of this paper is organized as follows: section 2 describes the

study area and the datasets used to train and test FarmCan. Section 3 describes FarmCan model development,

performance, and validation of model results. Major conclusions of the study are presented in Section 4.


## 2  Materials and Methods

### 2.1  Study Area

The CPE region covers the southern portions of the Prairie Provinces (Alberta (AB), Saskatchewan (SK), and Manitoba (MB)). Over 80% of Canadian farms are concentrated in the CPE (Wheaton et al., 2005). This makes the region ideal for developing and testing robust crop NI methodologies. The climate of the CPE is predominately continental with long, cold winters, short, hot summers, and relatively low precipitation

amounts during the growing season. Average winter and summer temperatures are -10 °C and 15°C, respectively (Hadwen and Schaan, 2017). A total of 4 study sites, on average 160 ha each, were selected within the provinces of SK and MB (Fig1 and Table1). These farms were selected based on the fact that they are sites for other soil moisture core validation networks, such as the Agriculture and Agri-Food Canada (AAFC) RISMA network in Manitoba (Bhuiyan et al., 2018) along with the Kenaston Network in Saskatchewan for NASA

Soil Moisture Active Passive (SMAP) validation (Sadri et al., 2020; Tetlock et al., 2019). All of the 4 farms are rainfed and have alternating crop years (ECCC, 2013). To avoid farrow and water-logged conditions in spring, farmers use pasture, spring wheat, shrubland, and other cover crops. Planting typically occurs in late April and early May depending on field and weather conditions. Throughout this study, we consider a fixed 7-month window for the growing season: from April 1 to October 31.

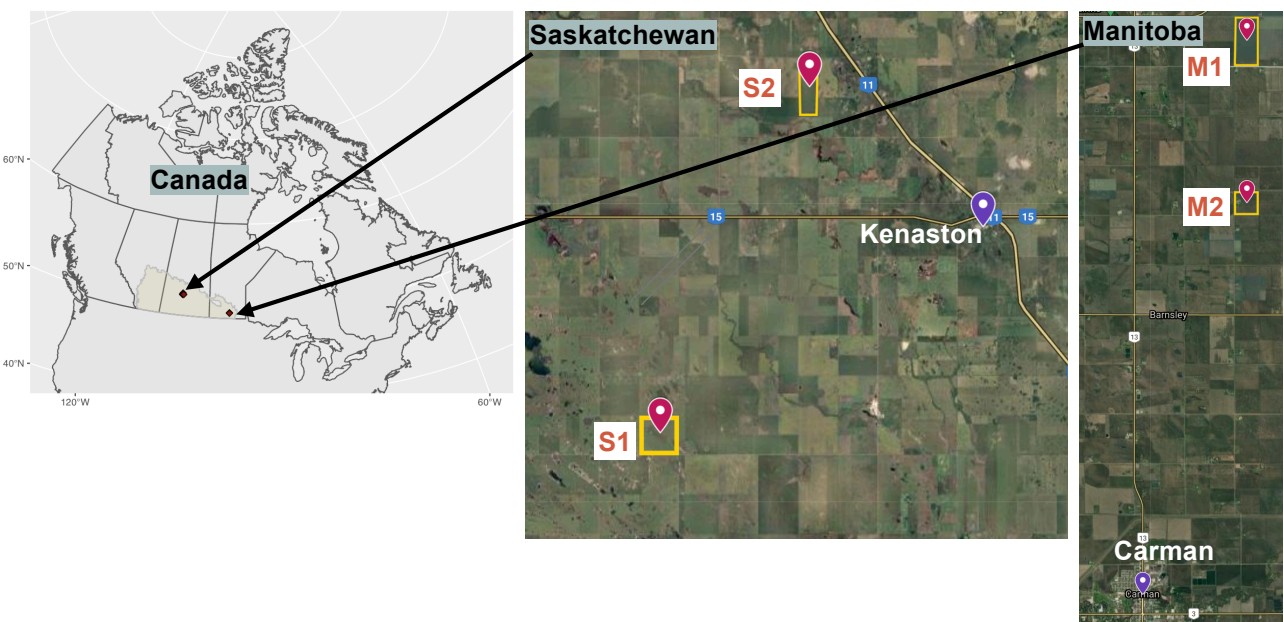

**Figure 1.** Location of 4 study farms in Saskatchewan (S1 and S2) near Kenaston, and in Manitoba (M1 and M2) near Carman (©Google Earth 2021).



**Table 1.** Information about each of the 4 study site including land use from 2015 to 2019.

| Study site | Lat | Long | Area [ha] | Dominant Land Use [total mm crop water need during growing period] | | | | | P[mm] Apr1–Oct31 | P[mm] Annual | P/PET [mm/day] |
| | | | | 2015 | 2016 | 2017 | 2018 | 2019 | | | |
|---|---|---|---|---|---|---|---|---|---|---|---|
| S1 | 51.42335 | -106.46100 | 263 | lentils [325] | canola | barley[450-650] | canola | spring wheat[450-650] | 122.45 | 146 | 0.292 |
| S2 | 51.55185 | -106.37318 | 192 | spring wheat | canola | peas[350-500] | spring wheat | canola | 131.62 | 155 | 0.282 |
| M1 | 49.67328 | -97.95417 | 130 | canola | spring wheat | soybeans[450-700] | canola | spring wheat | 167.6 | 211.7 | 0.385 |
| M2 | 49.62460 | -97.95435 | 65 | oats[450-650] | soybeans | oats | soybeans | oats | 179 | 221 | 0.402 |

\* P/PET: aridity index
\*\*PET is obtained from National Atlas of Canada in this stage





Table 1 shows that between 2015 to 2019, at least 7 different crops were planted in the 4 study farms. The majority of crops were canola and spring wheat, although there were also soybeans, oats, barley, peas, and lentils. All of these crops have low to medium sensitivity to drought and their root depth at maximum growth is anywhere from 0.6 m (lentils and soybeans) to 1.5 m (canola, barley, and spring wheat). The average crop water needs through the total growing season is 550 mm; much less was provided by rain (Shuval and Dweik,

2007; Brouwer and Heibloem, 1986). Table 1 also shows the amount of precipitation, mostly as snow, outside of the growing season. Establishing soil-water reservoirs or having stubble fields (Pomeroy et al., 1990) can improve snow contribution to SM in the future, and consequently, can be incorporated into the FarmCan model.

In addition, comparing PET with the total annual precipitation confirms that the amount of water supplied

by precipitation is insufficient to meet optimum crop growth. The growing season aridity index (P/PET) of each farm is shown in the last column of Table 1. This index is used across the globe to represent the biogeographical distribution of vegetation and to estimate crop yield (Franz et al., 2020). Based on the aridity index, Manitoba farms have a higher expected crop yield than those in Saskatchewan. Such contribution of melting snow toward meeting future crop water requirements is not substantial, and for that reason, is not

considered in the FarmCan model.

### 2.2    Model Components

Needed Irrigation (NI) (sometimes called Irrigation Consumptive Water Use, ICU) is the volume of water needed to compensate for the deficit between PET as a demand factor, and P (over the crop growing period) and change in soil moisture content ($\Delta SM$) as supply factors (FAO, 2021).

$$NI \approx \sum P - \sum PET + \Delta SM \qquad (1)$$

Both PET and SM are key climate variables that link the water, energy, and the carbon cycle (Fisher et al., 2017; Entekhabi et al., 1996).

The two main requirements for the datasets used to develop FarmCan are: 1) they are collected since 2015 or earlier; and 2) they are available and accessible in NRT. These two factors would allow the FarmCan

algorithm to update automatically on a daily basis and be responsive to the users' inquiries. Note that various





other datasets were considered, such as topography, Leaf Area Index (LAI), and the ET product from the NASA ECOSTRESS satellite, but they did not meet one or both requirements for FarmCan. SMAP SM products, on the other hand, are available/accessible in NRT and provide a highly accurate descriptor of crops such as canola stress across the CPE region (White et al., 2020). SM is a direct measure of agricultural

drought (Sadri et al., 2020; Vergopolan et al., 2021). Reduced precipitation amount affects SM available for crops consequently reducing crop yield, inflicting enormous economic impacts in developed countries, and suffering by millions of people in less developed regions of the world (Sadri et al., 2018; Sheffield and Wood, 2007; Wilhite et al., 2007; EM-DAT, 2020). RZSM becomes important during particular growth stages (mid-season and late season) and affects crop growth at maturity stages, as well as final crop yield(Smilovic

et al., 2019). The inclusion of SM as a dynamic parameter within numerical modeling has improved forecast capabilities for hydrological and meteorological models(Tetlock et al., 2019; Wanders et al., 2014; Koster et al., 2009). For FarmCan, therefore, we used SM in lieu of LAI. Other NRT variables used are PET, ET, and P from a combination of satellite-observed and modelled variables, each of which have been identified as key predictors of crop water stress (Pendergrass et al., 2020; Brust et al., 2021).

The list of all the datasets used in this study is summarized in table 2. All input variables were clipped to the CPE domain.

**Table 2.** Datasets and the periods used to train and run the model in this study.

| Variable | Dataset | Source | Depth (cm) | Period | Gridded res. (km) | Temporal. res. | Reference |
|---|---|---|---|---|---|---|---|
| SM | SMAP Level-3 (SPL3SMP) | RS* | 5 | 2015/03/31-2020/12/30 | 36 | Every 3-4 days | (Entekhabi et al., 2014) |
| RZSM | SMAP Level-4 (SPL4MAU) | Assimilated model | 100 | 2015/03/31 2020/12/30 | 9 | Daily | (Reichle et al., 2019) |
| P | MSWEP V280 | Assimilated in-situ and model | 5 | 1979/01/01-2020/12/30 | 5 | Daily | (Beck et al., 2019) |
| ET | MODIS | RS | - | 2015/01/01-2018/12/30 | 0.5 | Every 8-day** | (Running, 2001) |
| PET | MODIS | RS | - | 2015/01/01-2020/12/30 | 0.5 | Every 8-day** | (Running, 2001) |

\* RS: Remote Sensing
\*\*8-day composite values



SMAP Level 3 soil moisture (0-5cm) (SPL3SMP) is a composite based on daily passive radiometer retrievals of global land SM in the top 5 cm of the soil that is resampled to a global, cylindrical ~36 km Equal-Area Scalable Earth Grid, Version 2.0 (EASE-Grid 2.0). For this study, we use version 4 of SPL3SMP

retrievals from the morning overpasses, in order to minimize uncertainties and bias from the in-situ data (Al Bitar et al., 2017). Because the SMAP satellite was launched in 2015, SMAP data are only available from 31 March 2015, to present.

The SMAP Level 4 (SPL4SMAU) data provide continuous, daily global RZSM estimates (0-1m) by assimilating low frequency (L-band) microwave brightness temperature observations (for which SPL3SMP is

the gridded version) into the GEOS-5 Catchment land Surface Model (CLSM) (Reichle, 2017; Reichle et al., 2015; Sadri et al., 2018), which is driven by surface meteorological data from the NASA Goddard Earth Observation System (GEOS) weather analysis (Brust et al., 2021; Rienecker et al., 2008). Additional corrections are applied using gauge- and satellite-based estimates of precipitation that are downscaled to the temporal and 9 km scale of the model using the disaggregation methods described in (Liu et al., 2011) and (Reichle

et al., 2011).

ET and PET data are derived from MODIS, a modified MOD16A2/A3 Terra Version 6 (Running, 2001) ET/Latent Heat Flux algorithm. MODIS ET is an 8-day composite dataset produced at 500 meters (m) pixel resolution globally. The units are 0.1 kg/m$^2$/8day (i.e. 0.1 mm/8day) which is the summation of total daily ET through 8 days. 0.1 is the scale factor meaning that the data had to be corrected by multiplying them by 0.1

(Running et al., 2019). The last acquisition period of each year is a 5 or 6-day composite period, depending on the year. The algorithm used for the MOD16 data product collection is based on the Penman-Monteith equation, which includes inputs of daily meteorological reanalysis data along with MODIS remotely sensed data products such as vegetation property dynamics, albedo, and land cover. Provided in the MOD16A2 v006 product are layers for composited ET and PET along with a quality control layer from 2001-01-01 to present.

MODIS data are available from 2010 to present.

Daily precipitation (P) from Multi-Source Weighted-Ensemble Precipitation (MSWEP) Version V220 was used as the meteorological input for FarmCan. MSWEP is a global P product with a 3-hourly 0.1° resolution covering the period 1979 to the near present (Beck et al., 2019). The product blends gauge-, satellite-, and (re)analysis-based P estimates to improve the accuracy of the estimates globally. MSWEP Version 1 (0.25°

spatial resolution) was released in May 2016, and since then has been applied regionally and globally for



modeling SM and ET (Beck et al., 2019; Martens and Coauthors, 2017), estimating plant rooting depth (Yang et al., 2016), evaluating root-zone soil moisture patterns (Zohaib et al., 2017), evaluating climatic controls on vegetation (Papagiannopoulou et al., 2017), and analyzing diurnal variations in rainfall (L. Chen and Dirmeyer, 2017), and various other applications (Beck et al., 2019). MSWEP V280 is largely consistent with

MSWX-Mid and -Long forecasts. Here, we used only the MSWEP V280 product for the purpose of building the FarmCan algorithm, however, for the future software development application, we will use the MSWEP V280 combined with the MSWX product (**?**), which provide forecasts.

For validation, we performed a spatial and temporal generalization test for understanding the ability of the FarmCan in training and prediction on all days of crop planting in 2020 and for all of the 4 study farms using

$R^2$, RMSE, and KGE parametric tests.

A unique aspect of this work is that it takes advantage of specific farm characteristics that can be provided by the farmer. These inputs are used as static data and match crop phenology, based on FAO guidelines, with the relevant depth of soil moisture in real-time. These additional model inputs include the coordinates of the farm, the crop type, the crop date from sowing, the total number of expected growing days, and the day of the

year the forecast is made for.

### 2.3   Model Structure

Figure 2 summarizes the design of the main steps for the FarmCan algorithm. The steps include:

1.  User inputs the coordinates of a farm, crop type, planting date, and total growing days.

2.  The algorithm locates the farm and calculates the dates of each of the 4 phenological stages of crop

180         growth.

3.  Gridded RS Data (i.e. P, SM, RZSM, ET, PET, P) are clipped from the main datasets within calculated radii from the farm center in a way that each radius for each variable includes the closest-distance gridded data surrounding the farm. After averaging the data, each farm has one time series for each variable. Data are further processed for the 8-day composite or change values and are saved into the

185         system. This can accelerate the process of calculations for the same farm after the first round of data retrievals and processing.



4. Using RF, we fill in the missing values of the $\Delta$RZSM and $\Delta$SM so their data can be extended to 2010. By extending the soil moisture variable to a longer period, we make it consistent with the ET and PET record length which also provides a better prediction quality of FarmCan.

5. Composite 8-day P is used as the predictor and since $\Delta$RZSM has the highest correlation with 8-day P, it is the first predictant used with 8-day P in RF to forecast up to two weeks of $\Delta$RZSM. The predicted $\Delta$RZSM will then be jointly used with the 8-day P as predictors in the next step to predict $\Delta$SM. The process repeats and in every step, a new variable first is predicted and then used as a predictor.

6. Using equation 1, 8-day NI is calculated. We distributed the composite 8-day NI for the 8 past days
by calculating adjusted daily weights ($w_{adj}$) and deficit precipitation $P_{deficit}$ in a way that days with more P get less portion of the NI and vice versa. $P_{deficit}$ is the daily percentage of missing rain from the 8-day P. For example, a day $i$ with no precipitation has $P_{deficit}$=100% of the 8-day P and a day with 45% of 8-day P, has a $P_{deficit}$=55%. Daily $w_{adj}$ (in %) can then be calculated as:

$$w^i_{adj} = \frac{(800 - \Sigma^8_{i=1} P^i_{deficit})}{8} + P^i_{deficit} \qquad (2)$$

where 800 is the total deficit percentage in the absence of no rain in 8-days ($\sum^i_{w_{adj}} = 800\%$), $w_{adj}$ is then multiplied by the average daily NI to find the daily supplementary water amount. The sum of daily average NI should be the equivalent to 8-day NI.

## 2.4 Random Forest (RF) algorithm

RF was selected based on its promising capabilities in function estimation and high accuracy in nonparametric
regression in geospatial hydroclimatic and spaceborne data, even in the presence of collinearity (Clewley et al., 2017; Vergopolan et al., 2021). RF is a method for ML and is used for classification or regression problems. The RF algorithm works by aggregating the predictions made by multiple decision trees of varying subsets. Every decision tree in the forest is trained on a subset of the dataset called the bagged or bootstrapped dataset. Bagged sampling is a way of de-correlating the trees by showing them different training sets (Sonth
et al., 2020). This also decreases the variance of the model without increasing the bias which ultimately leads





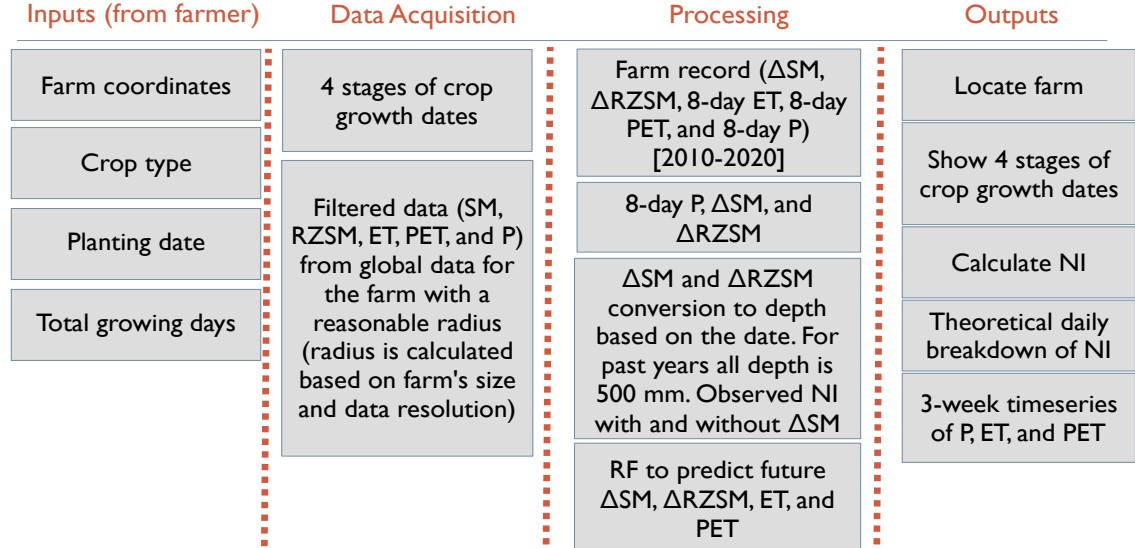

**Figure 2.** A chart description of the structure of FarmCan.

to better model performance. This means that while the predictions of a single tree are highly sensitive to noise in its training set, the average of many trees is not, as long as the trees are not correlated.

If RF is used for classification and is presented with a new sample, the final prediction is made by taking the majority of the predictions made by each individual decision tree in the forest. If RF is used for regression and is presented with a new sample, the final prediction is made by taking the average of the predictions made by the individual decision trees in the forest. For example, if a training set $X = x_1, ... x_n$ ($n$ being the number of training samples) has responses $Y = y_1, ... y_n$, the algorithm selects random samples with replacement of the training set for B times. For $b = 1, ... B$ training samples from $X, Y$, called $X_b, Y_b$, produce a regression tree $f_b$. After training, predictions for unseen samples $x'$ can be made by averaging the predictions from all the individual regression trees.

$$\hat{f} = \frac{1}{B} \sum_{b=1}^{B} f_b(x')$$
(3)





FarmCan model uses RF in two stages: 1) to fill in the gaps of missing $\Delta$SM and $\Delta$RZSM based on P from 2010 to 2020; and 2) to predict $\Delta$RZSM, $\Delta$SM, 8-day ET, and 8-day PET up to 14 days in advance. The datasets from 2010 to 2020 were divided into training and testing in a 0.7 to 0.3 ratio. The first round of

running RF uses 500 decision trees. The optimum number of trees is the one that minimizes the MSE between the training and testing datasets. The second round of running RF, involves dictating the optimum number of trees.

### 2.5    Relative importance of FarmCan inputs to P

At its core, FarmCan is based on the P as it is the only product with an available predicted value. Other

variables (ET, PET, SM, and RZSM) are used first as predictants and then as predictors to provide the model with antecedent information to ultimately produce NI forecasts. In order to decide on the order of variables being used with P in RF, we determined the relative importance of the other 4 variables in relation to P. We did so by running a two-by-two Pearson correlation analysis of variables (Figure 3) with 99% significance level for the 4 selected farms and for the 7-month periods of growing seasons from 2015 to 2020 based on

8-day temporal resolution. The correlation plots in Figure 3 are color-coded to show the significance of the correlation level. The analysis for all 4 farms' cumulative 8-day P showed similar patterns. $\Delta$RZSM and $\Delta$SM, respectively showed the highest correlation with 8-day P. The 8-day P is less correlated with the 8-day ET and PET. Here we do not see significant feedback from ET to SM and vice versa. This can be due to the fact that the relationship between SM and ET, in terms of feedback, mostly depends on the climate of the

location. During the growing season, the condition in CPE is either too wet, which makes the total energy for ET independent of soil moisture; or too dry, which makes ET show little impact on fluxes because there is little or no moisture available (I.Seneviratne et al., 2010).

Generally, a significant impact of SM on ET should be more noticeable in a transitional regime, where soil water supply is available and sufficient (Yang et al., 2020; Running et al., 2019; I.Seneviratne et al., 2010;

Famiglietti and Wood, 1994).

$\Delta$ SM or $\Delta$ RZSM volumetric values in the equation 1 were converted into their equivalent depth by multiplying them by the equivalent depth of the soil (mm) (Allen et al., 1998a, b). We make the assumption that soil moisture is distributed evenly across the depth. For example, a $0.2\text{m}^3/\text{m}^3$ of the surface SM (in the first 50 mm of the topsoil) is equivalent to: $0.2 \times 50 = 10\,\text{mm/day}$, whereas the same volumetric soil





moisture for root zone (with consistent depth of 1000 mm) is equivalent to $0.5 \times 1000 = 200\,\mathrm{mm/day}$. That

is, hypothetically, 200 mm of water can be drawn from 1 m deep soil. In our algorithm, the choice of using

50 mm or 1000 mm depth depends on the crop's development stage. When the crop is in stages 1 or 2, the

algorithm uses the first 50 mm depth, and when the crop is in stage 3 or 4, the 1000 mm depth is used.

**Figure 3.** Relationship between ET, PET, P, SM and SM changes for farms S1, S2, M1, and M2.





## 3 Results

### 3.1 Climatology comparison of hydrological variables

Figure 4 shows the climatology of the variables used in this study at 20th, 50th, and 80th percentiles from 2015 to 2020 during the growing season (April to October). From Figure 4, growing season precipitation ranges from 400-1200 mm. This amount of rain is typical of sub-humid and semi-arid climates (Allen et al., 1998a), i.e., that amount of rainfall is often not sufficient to satisfy the water needs of crops. With the exception of 260 portions of the province of AB, the majority of CPE farming relies on rainfall and therefore, is vulnerable to agricultural drought (Maybank et al., 1995; McGinn and Shepherd, 2003; White et al., 2020). Consequently, the yields in the CPE are expected to be less than optimal. Comparing precipitation with the ET and PET map shows that, region-wide, crops do not receive the water needed from rain to reach an optimal yield.

Most of MB and northern AB receive the highest amount of precipitation, so their need for supplementary 265 water to cover the water needs of the various crops is less relative to the CPE. Surface soil moisture is generally lower than RZSM across all three provinces. This might be due to the soil type (brown-black chernozemic clay) that holds the water for longer times. Most of Saskatchewan is identified by the lowest amount of soil moisture, precipitation, and ET throughout the growing season.

### 3.2 The feedback from a supply-demand mechanism

To understand the relationship between water supply and demand in the CPE we compared the variability among 8-day ET, 8-day PET, 8-day P, $\Delta$SM, and $\Delta$RZSM and showed the results in Figure 5. In this experiment, we also analyzed each province separately to see if there are significant differences among the variables of each province. A color gradient shows the changes of a 3rd variable which is the demand variable. For each province, the two left plots show the linear relationship between 8-day P and $\Delta$ SM, however, one changes 275 color with 8-day PET and the other with 8-day ET. While the organization of the points remains the same in the two left plots the 8-day PET is constantly higher than the 8-day ET. The same is true for the two right plots, except that the supply changes indicate 8-day P and $\Delta$ RZSM. This confirms that at the CPE a higher than supplied atmospheric demand exists throughout the growing season. For all three provinces, there is a stronger linear relationship between 8-day P and $\Delta$ RZSM than between 8-day P and $\Delta$ SM, especially

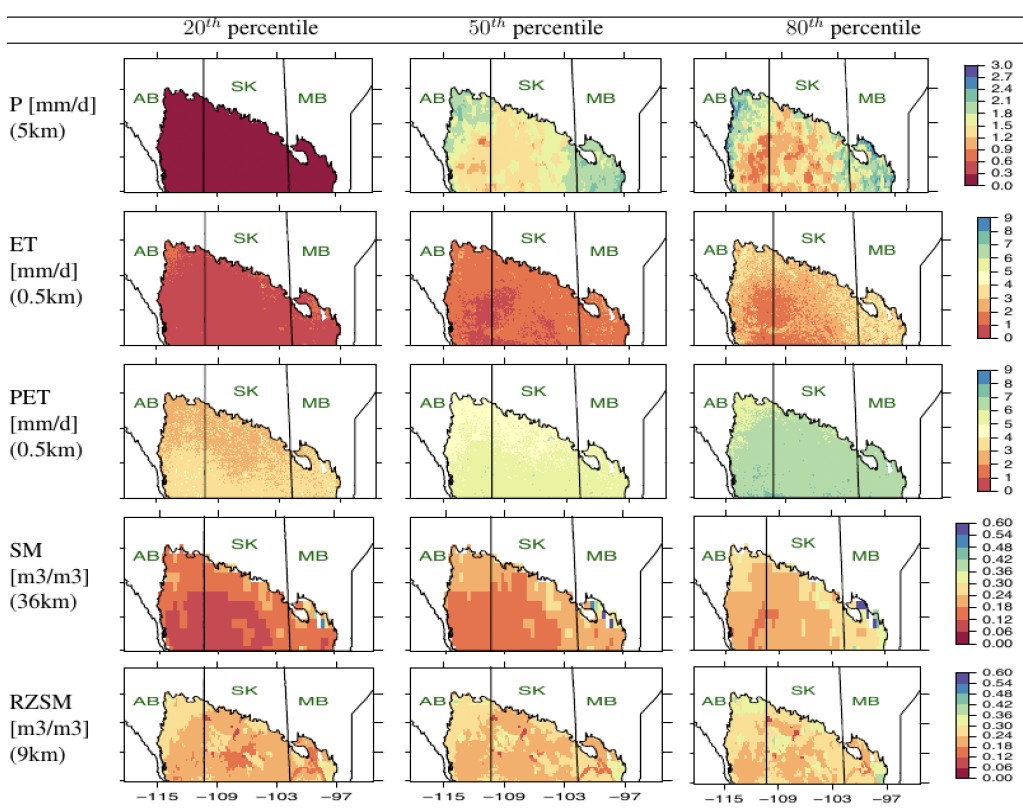

**Figure 4.** Spatial patterns of climatology. Data was collected from 2015-2020 for the agricultural months (Apr-Oct).

for Manitoba. Alberta has the weakest linear relationship between the supply variables perhaps due to being heavily affected by irrigation. Both Saskatchewan and Alberta have more 8-day periods when the 8-day PET is above 50 mm even though the 8-day P and the $\Delta$ RZSM are at their maximum values. This can be due to the fact that during the growing season, higher amounts of rainfall occur, or are followed by warmer days with little cloud coverage. For 8-day periods with less P, the atmospheric demand can be low or high

but generally, higher 8-day PET corresponds with a decrease in RZSM and SM (negative $\Delta$ values). Due to





relatively negligible differences among the patterns observed in the three provinces in the scope of the study, we do not further analyze the provinces separately.

**Figure 5.** Scatter plots of changes of Δ SM and 8-day P (supply) with 8-day ET and 8-day PET (demand). Each row shows one province. Data was collected from 2015-2020 for the agricultural year (months of Apr-Oct).





### 3.3 Historic data and calibration period

Figure 6 is the variability plot of farm S2 during the 7-month agricultural period (shown as pink background)

during 2015-2020. A negative $\Delta$RZSM or $\Delta$SM means a decrease in SM or RZSM, respectively, over the past 8 days and a positive $\Delta$RZSM or $\Delta$SM means an increase in SM or RZSM over the past 8 days. Changes of $\Delta$SM with time are more drastic than those for $\Delta$RZSM. Furthermore, the 8-day PET is consistently higher than 8-day ET suggesting, once more, that crops in farm S2 receive less than the optimal amount of their water demand throughout the year. We plotted variability plots for the other three farms (not shown here) and the

patterns were consistent with that of farm S2.

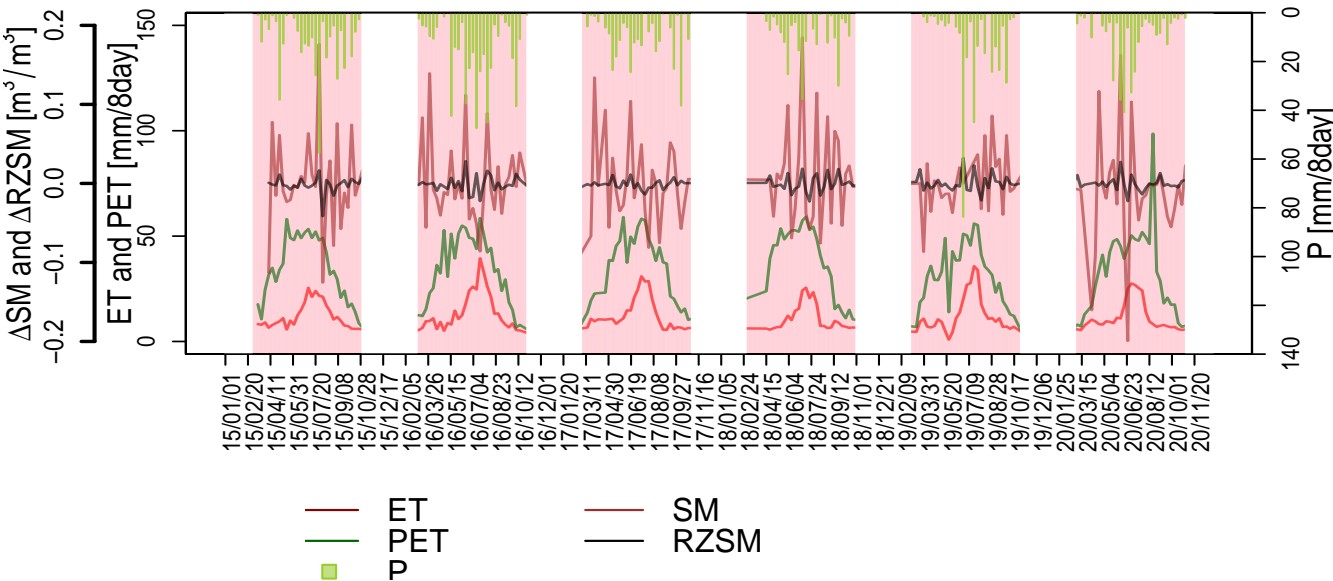

**Figure 6.** 8-day variability analysis, Farm S2 [2015-2020]. Pink background indicates the agricultural period. Green: PET, Red: ET, Purple: $\Delta$SM, Balck: $\Delta$RZSM, and Teal: 8-day P.





### 3.4 FarmCan prediction process

To illustrate the FarmCan real-time forecast process, we describe an example where 2020/07/02 is "today's date", the crop type is barley, the planting date is 2020/04/01, and the total growing season during is 150 days. The past dates are also used for validation purposes. The FarmCan algorithm takes these inputs and uses FAO

guidelines to provide the expected dates of stages 1 to 4, as shown in Table 3.

| Stage | Stage ending date |
|---|---|
| 1: initial | 2020/04/15 |
| 2: crop development | 2020/05/14 |
| 3: mid season | 2020/07/17 |
| 4: late season | 2020/08/25 |

**Table 3.** Key dates relevant to barley planted on 2020/04/01. From FAO guidelines (Allen et al., 1998a).

For the assumed date, the available and observed variables are plotted in Figure 7(a). The total period shown in the plot is 21 days from 2020/06/22 to 2020/07/12. The green bars are the daily precipitation from MSWEP including the forecast values. The hindcast NI, shown by the grey bars, is distributed by calculation of $w_{adju}$. Because 2020/07/02 corresponds to the 3rd stage of crop development, FarmCan uses $\Delta$RZSM

data instead of $\Delta$SM for training and predictions. If today's date is in the middle of the 8-day observation, it is unavoidable that a few days before today's data do not have updated 8-day ET and 8-day PET. So after the RF predicts the 8-day ET and 8-day PET, the algorithm fills in the few days antecedent to today's date. NI (in mm) is calculated for the remaining number of days (here, 10 days) of the next two weeks shown in Figure 7(b). Forecast results for the rest of the study farms are shown in Figure 8.



(a) Available observed variables for farm S2 on a give day such as 2020/07/02.

(b) Farm S2, after prediction using RF.

**Figure 7.** Farm S2 before and after prediction relative to the date 2020/07/02. Over the next 10 days the total predicted PET = 66.51 mm, total predicted ET = 31.93 mm, total P = 30.46 mm and NI = 71.24 mm.





(a) Farm S1 predictions over the next 10 days: total predicted PET = 62.48 mm, total predicted ET = 32.69 mm, total P = 36.45 mm and NI = 52 mm.

(b) Farm M1 predictions over the next 10 days: total predicted PET = 69.63 mm, total predicted ET = 37.53 mm, total P = 18.36 mm and NI = 41.3 mm.

(c) Farm M2 predictions over the next 10 days: total predicted PET = 76.23 mm, total predicted ET = 4.46 mm, total P = 17.2 mm and NI = 43 mm.

**Figure 8.** (a) Example of predictions from farms S1, M1, and M2 for 2020/07/02 as "today's date".



## 3.5 Tool Validation

Figure 9 shows the $R^2$ and RMSE values between the testing and predicted values of NI and for all the study farms. The ability of the FarmCan model to generalize the spatial regions (farms) was assessed by comparing these values. Spatially and across all of the 4 farms of study, FarmCan showed the highest correlations between observed and predicted values of 8-day ET, 8-day PET, and 8-day NI, and the lowest RMSE for ΔRZSM and ΔSM values. FarmCan forecast performance in the value estimation of 8-day NI was slightly lower than the observed values. The high $R^2$ and high RMSE for 8-day NI values suggest that the amount of NI might be under-predicted in FarmCan although the temporal patterns of water deficiency are well captured by the model. To confirm this result further, Table 4 shows the KGE values of 8-day ET, 8-day PET, and 8-day NI for the four study farms.

**Table 4.** KGE values of different covariates for different farms.

| Farm | ET | PET | NI |
|------|------|------|------|
| S1 | 0.70 | 0.52 | 0.43 |
| S2 | 0.78 | 0.87 | 0.43 |
| M1 | 0.76 | 0.83 | 0.45 |
| M2 | 0.76 | 0.83 | 0.53 |

The KGE of soil moisture values showed that the FarmCan model was not well-trained for ΔSM and ΔRZSM. This can be improved in the future as more data are gathered. Further studies on the effect of soil moisture inputs on the FarmCan can help to improve the model. Generally, there is inherent uncertainty in FarmCan forecasts since we cannot know the true value of the water deficiency and other controlling factors that maximize the crop yield. Despite this shortcoming, FarmCan showed an effective prediction capability of NI and improvement of our understanding of NI and some of its main controlling factors in the CPE.

(a) Farm S1

(b) Farm S2

(c) Farm M1

(d) Farm M2

**Figure 9.** 8-day correlation and RMSE plot for agricultural period of Farms S1, S2, M1, and M2 [2020].



## 4 Conclusions

This study evaluated the use and benefits of NRT remotely sensed datasets and ML for the CPE region. Crop water deficit data, P, and SM can be available to the farmer through in-situ measurements and weather stations, but gathering information and analytics for future planning from these data remains a challenge.
We demonstrated the potential of managing sustainable productivity of land by timing and tuning of water available to the crop through the design of FarmCan, a parsimonious supply-demand crop water monitor and forecasting mechanism. The methodology uses NASA's NRT remote sensing data representing both atmospheric and soil properties coupled with the information from the farmer, water balance, and ML to generate crop NI up to 14 days in advance as well as the historic graphs of data for the farm. We succeeded
to quantify the relative importance of both ET and SM on understanding the predictive value for NI in the CPE. We found that, compared to the daily data, 8-day ET, 8-day PET, 8-day P, $\Delta$SM, and $\Delta$RZSM are more useful calculators for predicting NI. The phenological stage of the crop had a determinant factor in using $\Delta$SM or $\Delta$RZSM in the model as well as converting the volumetric values of SM to depth. $\Delta$RZSM and $\Delta$SM are the two variables that showed a strong correlation with 8-day P. The 8-day ET and 8-day PET
showed to be more effective predictors of NI and FarmCan forecasting ability. We saw very little impact on fluxes between ET and SM in the CPE during the agricultural year. We speculate that is due to the cropping season climate at the CPE that is too wet at the bagging and too dry in the middle to the end of the agricultural season. In other transitional climates, we expect to see that the total energy of ET is more dependent on SM. More studies are needed to understand such feedback in the CPE and elsewhere and how that would affect
the development of FarmCan in the future. We also showed that in the CPE optimum crop production in the dry season should only be possible with an extra supply of water, while crop production in the rainy season may be possible but unreliable. The future version of this product could use weather forecast data for driving this model such that the predicted NI is based on forecast rather than climatology. Although FarmCan presented the potential benefits of saving water for farmers while calculating the NI for limiting crop water
stress, decisions about how much water should be supplied will need to be made within a larger community dialogue within management goals.





FarmCan can become a promising tool to help users focus their investment decisions during prolonged periods of drought or waterlogged conditions, schedule cropping and fertilizer applications, and address policy issues.

Future developments will focus on the role of the water retention capacity of the soil and crop type as two important factors potentially affecting NI measurements. Plants in sandy soils, for example, may undergo water stress quickly under water-deficient conditions, whereas plants in deep soil of fine texture may have ample time to adjust to low moisture conditions and may remain unaffected by any water deficiencies. Cotton shows complex responses to water stress because of its deep root system and it has the ability to maintain

low leaf water potential. Future developments also will focus on addressing how farmers can access FarmCan data, how using supplementary irrigation versus rain-only farming can help farm cost/benefit management, and how the NI predictions and management advisory actually aid in better on-farm water management and crop yield. Coupling fertilization timing and amount is another direction that can benefit farmers, especially in the context of food and water security. As part of this effort, we will be developing an online tool to make the

information available and responsive to farmers' inquiries. In order to conduct yield and cost-benefit analysis from using supplementary irrigation versus rain-only farming, future work will focus on receiving feedback data from the farm managers.

Despite some shortcomings, FarmCan is a step toward providing knowledge that can assist farm managers to make better decisions about excess water needs, drainage requirements, and even timing and amount of

fertilizer consumption.

*Author contributions.* Sara Sadri: Conceptualization, Methodology, Software, Formal analysis, Writing original and final draft; Ming Pan: Resources, Methodology, Editing, Consulting; Aaron Berg: Editing, Consulting; Hylke Beck: Resources, Editing; James S. Famiglietti: Resources, Consulting, Editing; and Eric F. Wood: Conceptualization, Consulting

*Competing interests.* The authors declare that they have no known competing financial interests or personal relationships
that could have appeared to influence the work reported in this paper.



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
