# Peer review of "FarmCan: A Physical, Statistical, and Machine Learning Model to Forecast Crop Water Deficit for Farms"

_Hydrology and Earth System Sciences, 2022_

## Referee Comment (RC2)

**FarmCan: A Physical, Statistical, and Machine Learning Model to Forecast Crop Water Deficit at Farm Scales**

Sara Sadri1, James S. Famiglietti1, Ming Pan2, Hylke E. Beck3, Aaron Berg4, and Eric F. Wood†

[referee-annotated manuscript omitted]

---

## Editor Comment (EC1)

Dear authors,

This paper presents a study of using a machine learning framework, "FarmCan", to forecast irrigation demand in 4 farms in Canada using machine learning. The authors find that soil moisture shows a strong correlation with precipitation and that ET and PET are effective predictors of NI. The study shows the potential of using machine learning models to improve the timing of irrigation and therefore to save water and achieve sustainable agricultural production.

The Editors would like to acknowledge the efforts all of the reviewers who have made comments on this manuscript. Due to an editorial issue during the invitation of reviewers phase, we have received 12 reviewer comments on the submitted manuscript, and an additional two community comments (14 reports to address in total). Invitations for review were sent out to a large volume of reviewers and we had an unusually large number of request for manuscript reviews accepted. Although this highlights the novelty and importance of the work undertaken, it is unrealistic to expect you to address all of these comments in turn.

In light of the editorial issues, I recommend that the Authors reply individually to three Reviewer comments (RC 1, RC 2, RC 3), which request relatively minor revisions. Further, I recommend that the Authors also reply to this Editor comment addressing the specific and general comments which have been summarised and taken from the remaining reviewer comments below (divided into general and specific comments, as well as those relating to the figures/tables).

Generally, most reviewers recommend that a reviewed version should be accepted after minor revisions (with the exception of Reviewers 9 and 12; RC9 recommend work due to some methodological shortcomings, and RC12 recommend rejection of the paper as they do not recognise the added values for predicting future conditions).

Once again, we apologise for the unusually large amount of reviewer comments received, but we hope this solution helps in responding to all of the reviewer's comments whilst recognising the inputs from all reviewers. Thank you again for submitting your manuscript to this Special Issue in HESS and we look forward to receiving your response.

Kind regards,

Dr. Daniel Green

**General comments:**

**[RC12]** First, it is unclear for me if their statistical model has a significant added value for predicting future conditions. Since they found that there is no significant relationship between P and PET (and ET), I guess that the initial condition of ET and PET is the major source of predictability of them (although they did not clarify this point). In this case, the authors may be able to replace their prediction of ET and PET with the persistent model. I think Figure 7 also implies that the persistent model is effective to predict them and it is not absolutely necessary to predict the dynamic change in ET and PET. In addition, the temporal change in soil moisture is also important, but the skill of their model to predict soil moisture is not actually good according to Figure 9. Although the precipitation prediction significantly contributes to the prediction of needed irrigation through equation (1), precipitation prediction comes from the existing data and is not the contribution of this work. In summary, without more detailed comparisons between their prediction and some benchmarks such as a persistent model, I cannot be very convinced that the authors' statistical model really provides an added value.

Second, it is unclear for me how this work contributes to estimating crop water conditions at farm scales since they fully relied on satellite observation with coarse grid sizes. Specifically, the size of the original footprint of SMAP is approximately 50km, which is apparently not a farm scale. Although it might be possible to integrate local information into the authors' proposed framework, I could not find any contributions to farm-scale water resource management in the present work.

**[RC9]** There are already numerous tools available to predict water demand for crop management. The novelty of this study is for the user to select a specific farm location, which alone is not sufficient for publication. Therefore, the novelty of this study needs to be better explained.

**[RC4]** There are really a lot of acronyms: please list them with the correspondent explanations at the end. **[RC10]** There are too many abbreviations in the text, and many of them are given without explanation; and it makes reading the text difficult. I would suggest putting all notations into the table in the Annex.

**[RC7]** The model, for example, produces a KGE for fourteen-day predications of something like 0.4. The authors provide limited discussion of the implications of this performance. What level of uncertainty does that imply? What could be the social and economic costs (crop loss, reduced yield, water costs, etc.)? How does this prediction accuracy vary over the season? How does accuracy vary over the prediction horizon? These questions could be discussed qualitatively (in leaving work for later) or quantitatively (in trying to add more work here). The provide, either way, a more robust understanding of the utility of this model.

**[RC8]** In my opinion the term "Needed Irrigation" is not appropriately used in this paper and could be misleading. Since FarmCan has been applied to rainfed cropping system, I think that the term "water deficit" (as in the title) is more appropriate.

**[RC8]** The paper should also clarify how this tool could be practically employed in "near real time": what kind of strategies could be implemented "to minimize potential crop failure and losses" in rainfed cropping systems?

**Specific comments:**

**[RC9]** L14: "four" instead of "4"

**[RC9]** L16-18: This statement already indicates that the ML method was not sufficiently tested.

**[RC9]** L32-35: It is unclear why and to what extent irrigation demand forecasts are important to rainfed farmers. It is very unlikely that they would change their farm management just because irrigation demand forecasts are available. This statement is also not included in the citations.

**[RC4]** In the abstract you affirm "...our algorithm was able to forecast crop water requirements 14 days in advance…": I do not understand why in the rest of the paper (for example figure 7-8) the predictions are up to 10 days.

**[RC9]** L60: Use consistent capitalization

**[RC9]** L66: What do you mean by "subfield"? Please consider that the SMAP data has only 36 km resolution.

**[RC9]** L67: I don't understand why this method is tailored to this area, as the method seems to be generic.

**[RC10]** Lines 78-79: Please, provide the numbers while describing the climatology of the region (length of winter and summer seasons, [annual/growing period] amount of precipitation, relative humidity, etc.).

**[RC10]** Line 156: the abbreviation P was previously explained in the text, there is no need to repeat it here.

**[RC4]** Line 167: there is (?) - please correct it. **[RC9]** L167: There seems to be a citation missing.

**[RC12]** L165-167: This description is a bit ambiguous for me. I believe that the authors used the forecast of P. Please explicitly say the P prediction was used in this paper.

**[RC5]** The study area section needs to be more explanatory such as past climatic scenarios which will directly or indirectly affect the crop.

**[RC9]** L181-185: This procedure is not clear to me. How do you calculate the radii? How do account for the different spatial resolution of the different data sets? Are you averaging over the SMAP grid? How representative is the SMAP soil moisture for a specific field? [RC12] L181: "P" appears twice.

**[RC9]** L187-189: It is unclear how the extension of the SMAP data to 2010 was done in detail. Furthermore, it is likely that a machine learning method will lead to very uncertain estimates of soil moisture, and I therefore do not see its benefit for the predictive modelling. Either explain in more detail how and why the extension was done, or better leave it out.

**[RC9]** L190-192: From a viewpoint of a soil hydrologist, it is very strange and arbitrary to first predict RZSM and than use this for the prediction of SM, as SM should be much stronger controlled by P than RZSM due to infiltration processes. Please explain in more detail reasoning behind this. In addition, explain why you predicting SM at all?

**[RC9]** L195-202: Eq. 2 is not correct. To obtain the correct weights, the result from Eq. 2 must be divided by 800. Furthermore, this procedure is a strong simplification, as it does not distinguish on which day P fell within the 8 days, which makes a big difference in reality. Let us assume, for example, that 100 mm P fell on the first day of the week. In this case, the irrigation requirement for the crops for the following days would be much lower compared to if 100 mm P would fell on the last day.

**[RC10]** Line 255-256. I would suggest not using the term "climatology" when discussing statistics estimated from the 5-year period.

**[CC1]** It's a misprint in Line 250, "0.5x1000", I think the correct phrase is "0.2x1000".

**[RC9]** L284-285: But high PET values can be found also for positive delta values of RZSM and SM for all regions.

**[RC9]** L292-294: This statement contradicts the time series of soil moisture shown in Fig. 6. If it were true, one would expect soil moisture to decrease continuously during the growing season. However, the delta values of SM and RZSM both show fluctuations around zero, suggesting that P is sufficient to compensate for ET losses. This discrepancy may be due to uncertainties in the PET and ET data.

**[RC9]** L309: In Fig. 8, M1 and M2 show strong P events of about 30 mm on 1 July. This results in an increase of SM to about 0.5 m³/m³ indicating soil saturation. Nevertheless, the model predicts a decrease in ET which is not plausible.

**Figures and tables:**

**[RC4]** In general some figure and tables are not very clear. If the information in Table 1 are divide in two parts the two new figures fits horizontally and this improves the readability. Figure 6 contains a lot of information and the colour and the bars confuse the reader: maybe also here is possible to split the figure in two parts. Figure 7-8: I do not understand why you show the "observed variables for farmS2" separately and you don't do this for the other farms; maybe the "growth stage" (it is simply a line) is useless.

**[RC5]** Figure 1 should include the scale and North direction symbol.

**[RC5]** Figure 3 explains day 8, PET, ET etc. Why day 8 parameters are important and what about other days. Figure 6 also only describes the 8-day variability. What is the significance of the day 8 events? [This links back to the previous comment about 10 day predictors, versus 14 days in advance]

**[RC9]** Figure 4: I don't see the benefit for showing the spatial distribution of P, ET, PET, SM and RZSM as this study only concerns time series analysis. Instead, monthly climate diagrams would be very useful to better understand the climate and soil

hydrological situation in the four farms used in this study to develop and test the model.
**[RC4]** Figure 4: I would suggest adding the explanation of all abbreviations used in the plots. Also, please remove the term "climatology" from the notation to the figure.

**[RC9]** Figure 6: Colours for ET and SM not distinguishable.

**[RC10]** Figure 7: I would suggest rounding predicted values for PET, ET, P and NI: like P=30.46 mm -> P=30 mm since the precision of the modeled value never becomes better than the observed (measured) value. The precipitation is usually measured with the precision of 1 mm; if the precision of the observed precipitation is better than 1 mm, please provide the description of the measuring technique or the reference describing it.

**[RC9]** Figure 8: There should be no subtitles under the subplots. The plots are difficult to understand because of the large number of symbols. The meaning of the growth stage line is unclear.

**[RC9]** Figure 9: Font size is too small. NI is not an observed value.

**[RC9]** Table 1: The source for the crop water needs is missing and not all crops are covered. The values for P are much too low (in the text values between 400-1100 during the vegetation period are given). The values of P/PET seems to be too low as well. **[RC10]** I would suggest providing the numbers for the precipitation rounded to tens of mm (i.e 122.45 -> 122 ).

**References:**

**[RC9]** Reference: Taghvaeian, S., Andales, A. A., Allen, L. N., Kisekka, I., O'Shaughnessy, S. A., Porter, D. O., ... & Aguilar, J. (2020). Irrigation scheduling for agriculture in the United States: The progress made and the path forward.

**[CC2]** mentions some useful references which may also add to the discussion/context of the work undertaken and some of the key governing principles.

---

## Author Comment (AC1)

Dear Dr. Green,

First, we thank you for handling our manuscript, Hess-2022-96. We hereby respond to the reviewers' comments in green font below. We responded to the first three of the twelve reviews, as instructed. Additionally, we responded to the other reviewers' specific and general statements you summarized. The main changes are as follows:

1. added additional information to maps, showing the scale and north sign (Figure 1), broke down Table 1 into two tables, showed NI values in the pair correlation analysis (Figure 4), enhanced the font and clarity (Figure 5 and Figure 9), corrected the legends (Figure 6), and did the analysis again for Table 3 and Figure 9, and a list of acronyms at the end of the manuscript;

2. improved the explanations of several caveats and limitations;

3. added citations to several relevant studies;

4. clarified the equations and the results in a more consistent manner; and

5. improved the introduction of FarmCan and the contributions of the algorithm.

Sincerely,

Sara Sadri (on behalf of all co-authors)

**Review RC 1**

This paper presents a study of using a machine learning framework, FarmCan, to forecast irrigation demand in 4 farms in Canada. Based on the machine learning modeling results, the authors find that soil moisture shows a strong correlation with precipitation. Also, evaporation and potential evaporation are effective predictors of NI. The study shows the potential of using machine learning models to improve the timing of irrigation and therefore to save water and achieve sustainable agricultural production. The manuscript is on a topic of interest to the audience of HESS. I only have a few minor comments that I hope the authors could address in their revision.

Thank you for your comprehensive review of our manuscript.

Specific comments:

1. Lines 51-58: In this part, the authors could add a few more references and add more in-depth discussion about the current stage of ML models for irrigation water demand.

This section is reorganized and rewritten to address various issues, including adding more references for ML models.

"Over the past few decades, Machine Learning (ML) techniques have been progressively used to process large amounts of information created by remotely sensed data. Various machine learning algorithms, such as Random Forests (RFs), Support Vector Machines (SVMs), Artificial Neural Networks (ANNs), Genetic Algorithms (GAs), and ensemble learning, have been used on remote sensing information in farming {Virnodkar_S.S.-2020-01}. RF applications have become popular for addressing data overfitting, especially in geospatial classification and prediction of remote sensing data p{Vergopolan_N.-2021-01, Saini_R.-2018-01}. However, their application for evaluating crop water stress and NI using remote sensing data continues to be under-explored, and the existing methods can still be greatly improved {Virnodkar_S.S.-2020-01, Yang_Y.-01-2020, UIUC_2021-01}. {Poccas_I.-2017-01} used RF and SVM to model leaf water potential for assessing grapevine water stress. {Loggenberg_K.-2018-01} combined RF with remote sensing data to distinguish stressed and non-stressed Shiraz vines."

2. Line 101: I checked the citation (FAO, 2021), which has the equation as: ICU = ET – P – dS. Please revise equation (1).

Although it does not make a difference in the results and the calculations, I have changed the equation to match the citation (FAO, 2021).

3. Line 167: There is a question mark here, which I assume is a place holder for references.

Yes, that was an issue in the BibTeX file, which is now addressed!

4. Lines 171-175: This description suggests that the FarmCan model is site-specific. The authors could add some discussion here to explain the flexibility of the model. Also, the authors can add explanation how the model can be transferred to other farm fields.

FarmCan is versatile in being applied to any farm in the region. I clarified that in the text.

5. In Figure 6, I would suggest change the color scheme. It is a bit confusing with ET and SM both presented in reddish colors.

There was a mistake in the size and thickness of the legend for this figure. It is now fixed, and the colors are identifiable.

6. At the end of the result section, maybe the authors can add a subsection to discuss the practical application of the FarmCan model. For example, how can we use the model to improve agricultural water use management?

I edited the text in various spots to address this issue, especially in the Introduction. For example:

"In irrigated farms, information on NI can help regulate water deficit, achieve higher levels of crop produced per unit of water consumed, and optimize profit while minimizing potential negative environmental effects {Han-M.-2018-01, Chalmers_D.J.-1981-01, Taghavaeian_S.-2020-01}. However, information on the proper quantity of water to feed crops is also essential in rainfed areas with insufficient rainfall to maintain crop yields and soil conditions {Virnodkar_S.S.-2020-01}. As climate change and drought continue to impact crop water stress and food insecurity, rainfed farms in the U.S. and Canada are increasingly adopting irrigation technologies {USDA_2021-01}. For example, the Canadian Ministry of Agriculture is encouraging farmers in Saskatchewan to evaluate their potential NI and apply for irrigation development {Saskatchewan_2022-01}. Knowing the quantity and timings of the water supply gives farmers incentives for more efficient practices, allows them to identify the timing and amount of nutrients, and facilitates more extensive management planning and adaptation strategy goals {White_J.-2020-01, Levidowa_L.-2014-01, IPCC_2013-01, Geerts_S.-2009-01, Taghavaeian_S.-2020-01}."

**Review RC 2**

This is a short, informative and to my mind original, article on the development of a tool to improve grain farming in Canada. This topic is new to me in the reviews I have done and found I had a sharp learning experience to enlighten me.

Thank you for the positive comment.

There is nothing seriously poor herein that needs to be attended to. Most of my remarks are attached to the Figures and the odd Table, to make reading easier.

I broke down Table 1 into two tables to make the reading easier. I also corrected various aspects of the figures to ensure clear visual communication.

To repeat a passage I wrote as a comment after the conclusion, I make an appeal which I hope will help the readers of the article: "Most potential readers will probably scan the Abstract, look at the Figures and possibly read the Conclusion, before they decide to read the whole. Please

repeat the text referred to by the acronyms in this passage. Because you have a lot of them, please add them in an appendix for reference below the text. Your article is relatively short, so an extra page will not hurt!"

Yes, thank you for this comment. I have added a list of acronyms at the end of the manuscript.

After some tidying up, I recommend that a reviewed version will likely be acceptable to the Editor. I would be happy to see the revision.

We appreciate it. Thanks again for the review.

**RC 3**

Hydrological forecasts provide valuable information for agricultural planning and management. This paper has developed a physical, statistical and machine learning model, which is called FarmCan, to forecast crop water deficit at farm scales. One feature of FarmCan is the integration of remote sensing datasets, including soil moisture, root zone soil moisture, precipitation, evapotranspiration and potential evapotranspiration. Through the case study of four farms in Canada. The usefulness of FarmCan is demonstrated. There are three comments for further improvements of the paper.

Firstly, there is a gap between rainfed farms and needed irrigation. Specifically, four rainfed farms are investigated in this paper (Lines 85 to 86) and the attention is paid to the needed irrigation (Lines 107 to 112). It is noted that rainfed and irrigated systems are two distinct approaches to agricultural production and that irrigation is generally not involved in rainfed systems. Please clarify the issue of needed irrigation in rainfed farms.

Thank you for the thorough review. The study aims to understand whether rainfed farms' natural water supply is generally enough to meet a balanced crop water demand. NI is another name for Irrigation Consumptive Water Use (ICU) for optimized Water Use Efficiency (WUE). With climate change affecting the water supply of rainfed farms, the number of stakeholders now looking into adopting irrigation is rising. Therefore NI information is critical for both irrigated and rainfed farms. I added the explanation in the introduction:

"In irrigated farms, information on NI can help regulate water deficit, achieve higher levels of crop produced per unit of water consumed, and optimize profit while minimizing potential negative environmental effects {Han-M.-2018-01, Chalmers_D.J.-1981-01, Taghavaeian_S.-2020-01}. However, information on the proper quantity of water to feed crops is also essential in rainfed areas with insufficient rainfall to maintain crop yields and soil conditions {Virnodkar_S.S.-2020-01}. As climate change and drought continue to impact crop water stress and food insecurity, rainfed farms in the U.S. and Canada are increasingly adopting irrigation technologies {USDA_2021-01}. For example, the Canadian Ministry of Agriculture is encouraging farmers in Saskatchewan to evaluate their potential NI and apply for irrigation development {Saskatchewan_2022-01}. Knowing the quantity and timings of the water supply gives farmers incentives for more efficient practices, allows them to identify the timing and amount of nutrients, and facilitates more extensive management planning and adaptation strategy goals

{White_J.-2020-01, Levidowa_L.-2014-01, IPCC_2013-01, Geerts_S.-2009-01, Taghavaeian_S.-2020-01}."

This study is the first step toward planning farm management, such as scheduling fertilizer, crop yield assessment, and uncertainty analysis. Even in rainfed farms, we cannot achieve those goals without first knowing how much water is available in real-time and how it circulates about timing and crop schedule. Indeed, one can not manage what one cannot measure. I also mentioned that despite the findings of this study, decisions about how much water should be supplied would need to be made in a more extensive community dialogue within management goals.

Secondly, the irrigation if applied would augment soil moisture and then affect evaporation. In Eq. (1) on Page 7, the needed irrigation is calculated by using evaporation and soil moisture. The calculation seems to mix independent and dependent variables. Specifically, from the perspective of statistical modelling, if x depends on y then it may be improper to regress y against x.

This is a good point. However, it would be difficult to separate how long it takes for the soil to absorb a rain episode or when and at what rate the evaporation begins affecting that particular portion of the soil moisture. Therefore, to address the interaction delay among hydroclimatic factors, we are doing this analysis in 8-day composite periods to take care of those unseen delays among system components and to reduce errors.

Thirdly, the algorithm of FarmCan accounts for 4 phenological stages of crop growth (Lines 179 to 180). It is known that crop water requirements vary by the different stages even under the same background climate (https://www.sciencedirect.com/topics/agricultural-and-biological-sciences/crop-water-requirement).

That is a correct statement. However, the scope of the paper is not to count for crop water requirements at each stage. Instead, we use different stages as a benchmark for switching from surface soil moisture to root zone soil moisture to calculate available water for a given crop on the farm on a given day. We assume that the near-real-time PET is the atmospheric demand and indirect indicator of crop water requirement at any stage. Therefore the NI should provide a realistic estimate of the water missing relevant to the stage. Future analysis can also work on various tests to tweak, and bias corrects the NI based on any crop water demand stage.

In addition, the analysis involves multiple crops, including soybeans, oats, spring wheat, etc. Please illustrate how the different crops and crop growth stages are considered under the same framework of FarmCan. Given that there are numerous combinations of crops/stages, can the data presented in this paper provide enough samples to train the FarmCan? What are the sampling variability and parametric uncertainty for the FarmCan?

The FAO has a guideline for the total number of growth days for specific crops and the general length of each growth stage. The FAO guideline is incorporated in this study. The FamCan algorithm allows the user to select the crop type. The algorithm takes the user's crop type and the number of growing days and breaks it down into four stages. For now, major Canadian crops are

incorporated in the tool and can be selectable by the user. Change in crop type does not affect the sampling variability for the RF prediction model.

Below are a few minor comments:

1. Please add a flowchart of the steps of data processing and the dataset involved.

Figure 2 shows a simplified version of the flowchart to summarize the steps taken in each stage to clarify the process.

In Fig. 9, it seems the uncertainty ranges are determined by linear regression models. Can the FarmCan quantify the uncertainty by itself?

The FarmCan algorithm uses the observed data as they come in to calculate past weeks' uncertainty to update the training of the RF algorithm for more realistic next week's predictions. Figure 9 presents an overall evaluation of uncertainty for the algorithm. There are many ways to branch out this research. Focusing on uncertainty analyses and more detailed ongoing uncertainty reporting by FarmCan is undoubtedly one of those avenues we will explore in the future.

General comments:

[RC12] It is unclear to me how this work contributes to estimating crop water conditions at farm scales since they fully relied on satellite observation with coarse grid sizes. Specifically, the size of the original footprint of SMAP is approximately 50km, which is apparently not a farm scale. Although it might be possible to integrate local information into the authors' proposed framework, I could not find any contributions to farm-scale water resource management in the present work.

In response to this comment, we modified the introduction to better emphasize the added value of this study. The study develops an algorithm to estimate crop water availability in farms across the CPE. We intentionally use only near-real-time remote sensing data because such data are accessible everywhere and make the algorithm flexible to be used in other parts of the world where in-situ farm data cannot be provided. The SMAP L3 resolution is 36 km. In future research, it will be 9 km. Maybe the farm-scale term was confusing, but essentially, it refers to what information can be collected from remote sensing data. I removed the "farm-scale" wording to avoid confusion.

[RC9] There are already numerous tools available to predict water demand for crop management. The novelty of this study is for the user to select a specific farm location, which alone is not sufficient for publication. Therefore, the novelty of this study needs to be better explained.

In response to this comment, I modified the introduction to better emphasize the added value of this study. I expanded on remorse sensing and machine learning advances in crop water stress and NI evaluations. I also added what is remained to be challenging in this field. Also added several new

references to connect the reader to the previous studies. For example after explaining the remote sensing and ML advances, I wrote:

"Despite these advances, scientific NI evaluation methods have generally remained limited, with relatively low adoption by farmers. Some of the problems to date are:
\begin{enumerate}
\item Lack of access: many farmers across the globe do not have access to the results of NI models. Therefore, management practices mostly rely on farmers' experience rather than scientific NI models.
\item Lack of timely predictions: Producers need to make NI decisions several days in advance and require tools capable of accurately forecasting short-term crop water use.
\item Complex procedures: Many of these models have tenuous requirements for inputs, time, labor, and financial investment, making the model remain within the scientific domain and out of reach for potential users.
\end{enumerate}
Several studies have indicated that it will be highly significant to address plant water stress using ML, which will help farmers improve water and cropland management practices in the low water productivity areas, substantially enhancing the food security p{Virnodkar_S.S.-2020-01}. To improve crop water stress and NI deficit management focus should be on: (1) including short-term forecasts in NI schedulers, (2) reducing data, time, labor, and cost requirements for schedulers, (3) providing user-friendly decision support systems, and (4) incorporating remotely sensed data in scheduling p{Taghavaeian_S.-2020-01}."

"In this study, we develop the FarmCan model to address the abovementioned issues. FarmCan is a hybrid physical-statistical-ML model for NI scheduling and other agricultural applications. At its core, FarmCan is trained on surface soil moisture (SM), root zone soil moisture (RZSM), P, Evapotranspiration (ET), and Potential ET (PET) to monitor and forecast daily NI daily and up to 14 days in advance. The contributions of the FarmCan algorithm are to (1) use farm-specific NRT remote sensing data as inputs, (2) use ML to forecast PET, SM, and RZSM using P prediction, and (3) develop a climate-informed forecast of crop NI volume and its timing with up to 14 days lead time, (4) allow users to interact with the tool by finding their farms, choosing crop and growing days and getting on a plan that guides and inform them about NI through the growing season, (5) use both SM and RZSM depending on the timing and crop growth stage. Our analysis and framework are developed for the Canadian Prairies Ecozone (CPE) farms. Still, they can be transferred anywhere to inform farmers and other stakeholders where and when additional water is potentially needed to compensate for water deficits. The tool will provide valuable information to governments, water managers, agriculturalists, and industries' sustainable initiatives to grow more food and avoid waste with better-managed water."

[RC4] There are really a lot of acronyms: please list them with the correspondent explanations at the end. [RC10] There are too many abbreviations in the text, and many of them are given without explanation; and it makes reading the text difficult. I would suggest putting all notations into the table in the Annex.

Thanks for the feedback. I included a list of acronyms at the end of the manuscript.

[RC7] The model, for example, produces a KGE for fourteen-day predications of something like 0.4. The authors provide limited discussion of the implications of this performance. What level of uncertainty does that imply? What could be the social and economic costs (crop loss, reduced yield, water costs, etc.)? How does this prediction accuracy vary over the season? How does accuracy vary over the prediction horizon? These questions could be discussed qualitatively (in leaving work for later) or quantitatively (in trying to add more work here). The provide, either way, a more robust understanding of the utility of this model.

Thank you for this comment. Table 5 had to be revised after doing the calculations again. This improved the KGE values of the NI for the four farms. Additionally, after careful examination of the delta SM and delta RZSM, we think the KGE, Corr, and RMSE results were expected, primarily because of the small range of variability in the "delta" values. So we needed to revise this section. The following explanations were added to the text to discuss the implications:

"KGE test is the goodness of fit. Generally, values higher than 0.41 are considered reasonable and satisfactory model performance, but there has not been a direct reason to choose this benchmark across all models {Knoben_W.J.M.-2019-01}. We consider 0.5 < KGE satisfactory in this study. The model's goodness of fit is reasonable for ET, PET, and NI. The KGE values of delta SM and delta RZSM (not shown) have been zero or very close to zero. Here, KGE's negative values do not necessarily indicate a model that performed worse than the mean benchmark. The reason is that the range of delta values of SM and RZSM was relatively small (approx. [-0.87,0.03] m^3/m^3), making the values very sensitive to the statistical tests. For the same reasons, it was expected that delta SM and delta RZSM did not show a good correlation, although they showed the lowest RMSE values. Given the satisfactory performance in final NI calculations, delta SM and delta RZSM predictions did not negatively affect the model and NI."

The following was added to the conclusion part:

"We quantitatively showed that the rainfed farms in the CPE area do not get the water required for optimum crop growth. Climate change will further affect this situation, and farmers are encouraged to move toward water management and adaptation strategies. Future studies can focus on such water shortages' social and economic implications (crop loss, reduced yield, water costs). For daily predictions, we used RF using 3-week data from one week prior, the current week, and one week over, and the data from the same days in the past years. This functionality allows the FarmCan algorithm to automatically take care of the seasonal variability. In the next step, FarmCan will use the MSWX product, which enables this tool to function in real-time and as a prediction tool."

[RC8] In my opinion the term "Needed Irrigation" is not appropriately used in this paper and could be misleading. Since FarmCan has been applied to the rainfed cropping system, I think the term "water deficit" (as in the title) is more appropriate.

Thank you for this comment. I used this term to be consistent with the FAO terminology and the water balance equation in the manuscript. However, to address your comment, I clarified in the Introduction that NI knows the water quantity missing and a measure of crop water stress. Knowledge of NI helps enhance Water Use Efficiency even for rainfed farms. Specifically, I added:

"Needed Irrigation (NI) or Irrigation Consumptive Water Use (ICU) is the amount of water to reduce crop water stress, satisfy crop water demand, and enhance agricultural Water Use Efficiency (WUE) {FAO_2000-01}. In irrigated farms, information on NI can help regulate water deficit, achieve higher levels of crop produced per unit of water consumed, and optimize profit while minimizing potential negative environmental effects {Han-M.-2018-01, Chalmers_D.J.-1981-01, Taghavaeian_S.-2020-01}. However, information on the proper quantity of water to feed crops is also essential in rainfed areas with insufficient rainfall to maintain crop yields and soil conditions {Virnodkar_S.S.-2020-01}. As climate change and drought continue to impact crop water stress and food insecurity, rainfed farms in the U.S. and Canada are increasingly adopting irrigation technologies {USDA_2021-01}. For example, the Canadian Ministry of Agriculture is encouraging farmers in Saskatchewan to evaluate their potential NI and apply for irrigation development {Saskatchewan_2022-01}. Knowing the quantity and timings of the water supply gives farmers incentives for more efficient practices, allows them to identify the timing and amount of nutrients, and facilitates more extensive management planning and adaptation strategy goals {White_J.-2020-01, Levidowa_L.-2014-01, IPCC_2013-01, Geerts_S.-2009-01, Taghavaeian_S.-2020-01}."

[RC8] The paper should also clarify how this tool could be practically employed in "near real-time": what kind of strategies could be implemented "to minimize potential crop failure and losses" in rainfed cropping systems?

That is a good point. Farmers of rainfed farms worldwide lack adequate means to characterize crop water use. Thus agricultural water management often operates under conditions of unknown water deficiency. Our model quantifies such water deficiency regularly. There might not be immediately actionable solutions for water deficiency, but knowing how much water is missing facilitates more extensive community dialogue within management goals. Another benefit of this tool is enabling farmers to plan and time and amount of fertilizer applications to ensure maximum consumption by the plant and minimum soil contamination.

Specific comments:
[RC9] L14: "four" instead of "4"

Done.

[RC9] L16-18: This statement already indicates that the ML method was not

sufficiently tested.

I agree that the phrasing of the L16-18 statement was confusing. However, I tested my ML method in various stages and added the description below in the Results to better convey the meaning of the KGE tests:

"Here, KGE's negative values do not necessarily indicate a model that performed worse than the mean benchmark. The reason is that the range of delta values of SM and RZSM was relatively small (approx. [-0.87,0.03] m^3/m^3), making the values very sensitive to the statistical tests. For the same reasons, it was expected that delta SM and delta RZSM did not show a good correlation, although they showed the lowest RMSE values (Figure \ref{fig:validation}). Given the satisfactory performance in final NI calculations, delta SM and delta RZSM predictions did not negatively affect the model and NI."

[RC9] L32-35: It is unclear why and to what extent irrigation demand forecasts are important to rainfed farmers. It is very unlikely that they would change their farm management just because irrigation demand forecasts are available. This statement is also not included in the citations.

This is similar to another reviewer's comments above. To address this question, I edited the introduction in multiple spots. For example, I wrote:
"information on the proper quantity of water to feed crops is also essential in rainfed areas with insufficient rainfall to maintain crop yields and soil conditions {Virnodkar_S.S.-2020-01}. As climate change and drought continue to impact crop water stress and food insecurity, rainfed farms in the U.S. and Canada are increasingly adopting irrigation technologies {USDA_2021-01}. For example, the Canadian Ministry of Agriculture is encouraging farmers in Saskatchewan to evaluate their potential NI and apply for irrigation development {Saskatchewan_2022-01}. Knowing the quantity and timings of the water supply gives farmers incentives for more efficient practices, allows them to identify the timing and amount of nutrients, and facilitates more extensive management planning and adaptation strategy goals {White_J.-2020-01, Levidowa_L.-2014-01, IPCC_2013-01, Geerts_S.-2009-01, Taghavaeian_S.-2020-01}."

[RC4] In the abstract you affirm "...our algorithm was able to forecast crop water requirements 14 days in advance…": I do not understand why in the rest of the paper (for example figure 7-8) the predictions are up to 10 days.

You are correct. The algorithm can predict crop water demand "up to" 14 days in advance. This is because the algorithm functions daily, predicting the day remaining in the current week and the following week for each given date. If we are on the first day of the week, the algorithm predicts 14 days in advance, but if we are on the last day of the week, it predicts seven days in advance. I noticed the rest of the manuscript mentioned up to 14 days, except the abstract. So I corrected the wording in the abstract.

[RC9] L60: Use consistent capitalization

Not sure where this applies precisely. However, I carefully considered all the typos in this round.

[RC9] L66: What do you mean by "subfield"? Please consider that the SMAP data has only 36 km resolution.

Subfield in this context meant to show that low-resolution data were used to achieve NI for specific farms. I deleted the subfield word to avoid confusion since it was only mentioned once in the manuscript.

[RC9] L67: I don't understand why this method is tailored to this area, as the method seems to be generic.

The methodology is generic, however, due to computing storage limitations and also for the reason that this tool is in the initial stage of development, the approach was tailored (i.e., customized) in the CPE. I rephrased this in the paper.

"Our framework is tailored (i.e., customized) for the Canadian Prairies Ecozone (CPE). However, the methodology is generic and can be transferred anywhere to inform farmers and other stakeholders where and when additional water is potentially needed to compensate for water deficits."

[RC10] Lines 78-79: Please, provide the numbers while describing the climatology of the region (length of winter and summer seasons, [annual/growing period] amount of precipitation, relative humidity, etc.).

Done. The revised paragraph reads as follows:

"The climate of the CPE is predominately continental, with long, cold winters and short growing seasons of May to August {Bonsal_B.R.-1999-01}. The annual mean precipitation is around 478 mm, of which rainfall accounts for almost two-thirds of it during the growing season, and snowfall makes up another 30% of it. Average winter and summer temperatures are -10C and 15 C, respectively {AAFC_2017-01}. A total of 4 study sites, on average 160 ha each, were selected within the provinces of SK and MB\. These farms were selected based on the fact that they include some of the in-situ sites for SM core validation networks, such as the Agriculture and Agri-Food Canada (AAFC) RISMA network in Manitoba {Bhuiyan_H.A.K.M-2018-01} and the Kenaston Network in Saskatchewan for NASA Soil Moisture Active Passive (SMAP) validation {Sadri_S.-2020-01, Tetlock_E.-2019-01}. All four farms are rainfed and have alternating crop years {AnnualCropInventory_2020}. Farmers use spring wheat, shrubland, and other cover crops to avoid farrow and water-logged conditions in spring. Depending on field and weather conditions, planting typically occurs in late April and early May. This study considers a fixed 7-month window for the growing season: from April 1 to October 31."

.

[RC10] Line 156: the abbreviation P was previously explained in the text, there is no need to repeat it here.

Done.

[RC4] Line 167: there is (?) - please correct it. [RC9] L167: There seems to be a citation missing.

Done.

[RC12] L165-167: This description is a bit ambiguous for me. I believe that the authors used the forecast of P. Please explicitly say the P prediction was used in this Paper.

"MSWEP is a global P product with a 3-hourly 0.1-degree-resolution covering the period 1979 to the present. It does not provide a forecast. However, MSWEP V280 is largely consistent with a newer product, MSWX, that offers medium and longer-term forecasts. Here, we used past dates to build a forecasting tool, so using the MSWEP V280 product was sufficient. For future software development applications, we will use MSWEP combined with MSWX to provide real-time forecasts."

[RC5] The study area section needs to be more explanatory such as past climatic scenarios that will directly or indirectly affect the crop.

Thanks for the suggestions. The following paragraph was modified:

"Over 80% of Canadian farms are concentrated in the southern portions of the Canadian Prairies (Alberta (AB), Saskatchewan (SK), and Manitoba (MB)) called the CP Ecozone (CPE){Wheaton_2005-01}. The CPE has some of the world's highest climate and weather variability. It is predominately continental with long, cold winters, short, hot summers, and relatively low precipitation amounts during the short growing seasons of May to September {Bonsal_B.R.-1999-01}. The annual mean precipitation is around 478 mm, of which rainfall accounts for almost two-thirds of it during the growing season, and snowfall makes up another 30% of it. Average winter and summer temperatures are -10 C and 15C, respectively {AAFC_2017-01}. Such variabilities significantly affect CPE's agriculture, environment, economy, and culture yearly {Sadri_S.-2020-01}. For example, the drought of 2001–2002 cost approximately 3.6 billion in agricultural production losses {Wheaton_2005-01}. Between 2008 and 2012, federal-provincial disaster relief payouts for climate-related events totaled more than 785 million and more than 16.7 billion in crop insurance. The 100-year record-breaking drought in 2017 caused massive wildfires, reduced yields (particularly canola), heat stress, poor grain fill, livestock feed shortages, and relocation of nearly 3000 cattle in Saskatchewan and Alberta {Cherneski_P.-2018-01}. The vulnerability of the CPE to agricultural production risks and the future scenarios of climate, which show more severe and frequent droughts with declining

precipitation trends and surface water resources during summer and fall, makes the region ideal for developing and testing robust crop NI methodologies."

[RC9] L181-185: This procedure is not clear to me. How do you calculate the radii? How do account for the different spatial resolutions of the different data sets? Are you averaging over the SMAP grid? How representative is the SMAP soil moisture for a specific field?

The revised paragraph reads as follows:

"From the farm coordinates, the farm center is calculated. Gridded RS data (i.e., P, SM, RZSM, ET, PET) are clipped from the primary datasets using radii from the farm center calculated in a way that each radius for each variable includes the closest gridded data surrounding the farm perimeter. Calculations of the variables' radii are based on trial and error and the variable's spatial resolution. The farm's specific variable time series is filtered by interpolating the grids outside the perimeter and any of the grids inside the farm. Timeseries data are further processed for the 8-day composite or changed (delta) values."

[RC12] L181: "P" appears twice.

Thank you for pointing this out. It is fixed.

[RC9] L187-189: It is unclear how the extension of the SMAP data to 2010 was done in detail. Furthermore, a machine learning method will likely lead to very uncertain estimates of soil moisture, and I, therefore, do not see its benefit for predictive modeling. Either explain in more detail how and why the extension was done or better leave it out.

This part is essential as a way of gap-filling the data. However, I agree that more tests are required to ensure QA/QC is done on the gap-filled data. I removed this part to avoid confusion and deviating from the primary goal.

[RC9] L190-192: From a viewpoint of a soil hydrologist, it is very strange and arbitrary to first predict RZSM and than use this for the prediction of SM, as SM should be much stronger controlled by P than RZSM due to infiltration processes. Please explain in more detail the reasoning behind this. In addition, explain why you predicting SM at all?

There are a couple of points related to this matter:
1) First, this study has developed a methodology based on observations of 4 farms in one region (i.e., CPE).
2) The four farms' results show that the correlation between P and delta RZSM and P and SM is quite similar in M1 and M2. However, the correlation between P and delta RZSM is slightly higher in S1 and considerably higher in S2.

3) Although it makes sense that instantaneous surface soil moisture shows more variability with P, this study is based on the correlation between 8-day cumulative P and changes in 8-day soil moisture and not a direct measure of P vs. SM or SM RZSM. For example, if the total amount of P over eight days is 20 mm, the RZSM can change from 0.2 to .9 m^3/m^3, which gives a higher delta RZSM than delta SM, which might have fluctuated all the time but essentially changed from 0.3 to 0.5.
4) All said, more studies in different regions must confirm such correlations. Perhaps soil type can also play a role in which portion of the soil keeps the moisture for longer. I will add this explanation to the conclusion and in the text.

[RC9] L195-202: Eq. 2 is not correct. To obtain the correct weights, the result from Eq. 2 must be divided by 800. Furthermore, this procedure is a strong simplification, as it does not distinguish on which day P fell within the 8 days, which makes a big difference in reality. Let us assume, for example, that 100 mm P fell on the first day of the week. In this case, the irrigation requirement for the crops for the following days would be much lower compared to if 100 mm P would fall on the last day.

Thank you for mentioning this. Although I can confirm that the process and coding of the work were correct, I can see that the explanation in the manuscript may have confused the reader. Therefore, I added more equations and explanations to complete the calculation description.

[RC10] Line 255-256. I would suggest not using the term "climatology" when discussing statistics estimated from the 5-year period.

I agree that the period is not long enough to represent a stationary "climatology" with high confidence. But such plots are statistically referred to as climatology maps and five years is still informative. With the short-term SMAP data, for example, the 5-year percentiles maps are the best to present the variable patterns over a geographical region with the lower and upper percentiles. Of course, it might have a higher non-stationarity for climatology compared to maps from longer-term data. However, it still can be considered the climatology map that SMAP can show.

[CC1] It's a misprint in Line 250, "0.5x1000", I think the correct phrase is "0.2x1000".

Yes. Thank you for pointing that out. I fixed it.

[RC9] L284-285: But high PET values can be found also for positive delta values of RZSM and SM for all regions.

Yes, that is correct. Overall, I expanded upon the descriptions and explanations of the results in this section. The revised version is:

"To study the relationship between water supply and demand in the CPE, we conducted a 3-way comparison of changes in 8-day P supply with variability in delta SM and delta RZSM (supply).

We also included changes in 8-day ET and 8-day PET (demand factors) in a correlation plot shown in Figure \ref{fig:supply-demand}. Each row represents a province; the supply variables are on the XY axes. For each region, the two left plots show the relationship between 8-day P and delta SM. Color changes correspond with 8-day PET and 8-day ET. The two right-side plots are the same, except that the Y-axis represents delta RZSM instead of delta SM.

Manitoba shows the most robust linear relationship between 8-day P and delta RZSM. In contrast, Alberta shows the weakest linear relationship between 8-day P and delta RZSM, likely because most Alberta farms are artificially irrigated, making it hard to find linear patterns between P and SM responses. Our studies show that delta RZSM is more responsive to the amount of 8-day precipitation, meaning that over an 8-day increase of P, RZSM increases. However, such a linear relationship is weaker between 8-day P and delta SM. This can be because surface SM is also affected by exposure to other physiological elements such as wind, elevation, and land cover.

There are also linear relationships between the 8-day PET and 8-day P, especially in Manitoba and Saskatchewan. The 8-day PET (and less for 8-day ET) tend to increase with higher 8-day P. The 8-day ET and 8-day PET do not show a linear correlation to the changes in soil moisture, although for periods with 10 < 8-day P < 40 mm, 8-day PET values tend to be high when the soil moisture is decreasing.
When 8-day P > 40 mm, Saskatchewan and Alberta showed more mid-range PET (20-60 mm/8 day), whereas MB showed more swings. This means SK and AB are consistently dry. MB is generally moist but can range from adequate crop water availability to extreme stress. The atmospheric demand is typically low for periods with 8-day P less than 10 mm.
In all plots, the average 8-day PET is constantly higher than the 8-day ET. This confirms that a higher than supplied atmospheric demand exists throughout the growing season at the CPE."

[RC9] L292-294: This statement contradicts the time series of soil moisture shown in Fig. 6. If it were true, one would expect soil moisture to decrease continuously during the growing season. However, the delta values of SM and RZSM both show fluctuations around zero, suggesting that P is sufficient to compensate for ET losses. This discrepancy may be due to uncertainties in the PET and ET data.

It is unclear which statement contradicts Figure 6. If you mean "changes in delta SM is more drastic compared to changes at delta RZSM"..., I think Figure 6 shows precisely that the surface soil moisture reacts to P with much higher variability and sensitivity during every agricultural year than Delta RZSM. I revised the text to:

"During every agricultural year, the surface soil moisture reacts to P with much higher variability and sensitivity than delta RZSM. This makes sense because SM is affected by other physical elements such as wind and P, whereas RZSM is more related to soil type. In the CPE, delta RZSM generally reverts to zero, showing a weakly stationary behavior. However, the amount and timing of daily RZSM can still be insufficient to support sufficient crop growth. As for surface SM, the changes do not seem stationary. However, some years might seem more like a weakly stationary regime for most of the agricultural period. Therefore, more studies are required to

understand how the magnitude of delta and its timing affect crop development. For example, for the first month of cropping surface SM is greatly important. However, in most years, we see very low or very high changes in SM during the first month, suggesting a delayed cropping schedule or a compromised yield quality due to the poor initial seeding conditions.

The 8-day PET is consistently higher than 8-day ET, proving that crops receive less than the optimal amount of their water demand throughout the year. The gap between PET and ET was the most in 2017 and 2018. We plotted variability plots for the other three farms (not shown here), and the patterns were consistent with those from farm S2."

[RC4] In general some figure and tables are not very clear. If the information in Table 1 is divide in two parts the two new figures fits horizontally and this improves the readability. Figure 6 contains a lot of information and the colour and the bars confuse the reader: maybe also here is possible to split the figure in two parts. Figure 7-8: I do not understand why you show the "observed variables for farmS2" separately and you don't do this for the other farms; maybe the "growth stage" (it is simply a line) is useless.

I split Table 1 into two tables. In figures 7-8, I only showed the observed variables for farm S2 to establish the type of format of the observed data before RF prediction. The information in the first plot of S2 is repeated in the second plot of S2. I only included the resulting figures for the next farms to avoid redundancy. By looking at Figure 7, I believe the reader can identify the type of data already observed for the other farms. This way, I avoided redundancy by repeating the same information in Figure 8. I added an explanation for the reader in this regard.

"The total period shown in the plot is 21 days, from 2020/06/22 to 2020/07/12. The green bars are the daily precipitation from MSWEP, including the forecast values. The hindcast NI, shown by the grey bars, is distributed by calculating $w_{adju}$. Because 2020/07/02 corresponds to the 3rd stage of crop development, FarmCan uses delta RZSM data instead of delta SM for training and predictions. Therefore, after 8-day PET is predicted, the algorithm calculates 8-day NI (in mm) for the remaining days shown."

[RC5] Figure 1 should include the scale and North direction symbol.

Thank you. I added those to figure 1.

[RC5] Figure 3 explains day 8, PET, ET, etc. Why day 8 parameters are important and what about other days. Figure 6 also only describes the 8-day variability. What is the significance of the day 8 events? [This links back to the previous comment about 10-day predictors, versus 14 days in advance]

This has to do with how MODIS satellite data are collected. It is possible to design the algorithm for any internal that works using various statistical interpolating techniques. However, to avoid adding unnecessary biases to the process, we used the MODIS data as reported, which is an

8-day composite. We made P and SM consistent with the 8-day intervals, which was a more straightforward process than trying to break down ET daily. Our initial design tried to generate ET and PET on a 7-day basis from the 8-day composites, but after trying for a while, it seemed unnecessary to take that path.

[RC9] Figure 4: I don't see the benefit of showing the spatial distribution of P, ET, PET, SM and RZSM as this study only concerns time series analysis. Instead, monthly climate diagrams would be very useful to better understand the climate and soil hydrological situation in the four farms used in this study to develop and test the model.

I believe the plot provides valuable information about the big picture of our data. Understanding what these data look like statistically across the study region is essential for knowing that our farm-scale results are reasonable and within the range. Please note that I chose four farms for this study, but essentially we want any farmer in the region to get the data needed for their farm.

[RC4] Figure 4: I would suggest adding the explanation of all abbreviations used in the plots. Also, please remove the term "climatology" from the notation to the Figure.

I removed the word climatology and reworded the caption. I also added the explanation of the provinces that were abbreviated. Other variables are constantly repeated with the same name throughout the text.

[RC9] Figure 6: Colours for ET and SM not distinguishable.

Correct. It is fixed in the revised manuscript. We appreciate the comment.

[RC10] Figure 7: I would suggest rounding predicted values for PET, ET, P and NI: like P=30.46 mm -> P=30 mm since the precision of the modeled value never becomes better than the observed (measured) value. The precipitation is usually measured with the precision of 1 mm; if the precision of the observed precipitation is better than 1 mm, please provide the description of the measuring technique or the a reference describing it.

Good suggestion. I changed the precision in the caption. Thank you.

[RC9] Figure 8: There should be no subtitles under the subplots. The plots are difficult to understand because of the large number of symbols. The meaning of the growth stage line is unclear.

Based on other reviewers' feedback, I added more explanations throughout the text to clarify how the growth stage is calculated. I also removed the captions from Figure 8 and added a general caption at the bottom.

[RC9] Figure 9: Font size is too small. NI is not an observed value.

I regenerated the plot and made the labels bigger. NI is not directly observed but indirectly calculated from the observed values. I have mentioned this in the text.

[RC9] Table 1: The source for the crop water needs is missing and not all crops are covered. The values for P are much too low (in the text values between 400-1100 during the vegetation period are given). The values of P/PET seem to be too low as well.

I added the source (FAO) in the caption of the now Table 2. I made sure all crops were covered. Thank you for noticing the discrepancy between the table and the statement in the results. After recalculating, we can confirm that the information in the table is correct, but the numbering 400-1100 mm rain was indeed wrong. I appreciate your attention to detail. I corrected the text.

[RC10] I would suggest providing the numbers for the precipitation rounded to tens of mm (i.e 122.45 -> 122 ).

Done.

References:

[RC9] Reference: Taghvaeian, S., Andales, A. A., Allen, L. N., Kisekka, I., O'Shaughnessy, S. A., Porter, D. O., ... & Aguilar, J. (2020). Irrigation scheduling for agriculture in the United States: The progress made and the path forward.

Good article. Thanks for suggesting it. I included it in the revised manuscript.

[CC2] mentions some useful references which may also add to the discussion/context of the work undertaken and some of the key governing principles.

Agreed. I added several relevant references, including references from key governing organizations such as the Government of Canada and the FAO.

Thanks again for your thoughtful review.

---

## Author Comment (AC3)

This is a short, informative and to my mind original, article on the development of a tool to improve grain farming in Canada. This topic is new to me in the reviews I have done and found I had a sharp learning experience to enlighten me.

Thank you for the positive comment.

There is nothing seriously poor herein that needs to be attended to. Most of my remarks are attached to the Figures and the odd Table, to make reading easier. To repeat a passage I wrote as a comment after the conclusion, I make an appeal which I hope will help the readers of the article: "Most potential readers will probably scan the Abstract, look at the Figures and possibly read the Conclusion, before they decide to read the whole. Please repeat the text referred to by the acronyms in this passage. Because you have a lot of them, please add them in an appendix for reference below the text. Your article is relatively short, so an extra page will not hurt!"

Yes, thank you for this comment. I broke down Table 1 into two tables to make the reading easier. I also corrected various aspects of the figures to ensure clear visual communication.

I have added a list of acronyms at the end of the manuscript.

After some tidying up, I recommend that a reviewed version will likely be acceptable to the Editor. I would be happy to see the revision. My comments to the Authors follows my Signature below this passage, which is my wont.

We appreciate it. Thanks again for the review.

Geoff Pegram

6 May 2022

+++++++++++++++++++++++++++++++++++++++++++++++++++++++++++++++++++++++++++++++++++
++++

Details of comments inserted in the article.  Clips of your text are numbered and my remarks follow introduced by #.  I will not copy the trivial suggested corrections, but I will take the more pithy selections and add them here.

11 MSWEP

**Multi-Source Weighted-Ensemble Precipitation**

 In the new version, I completely rewrote the Abstract, so there is no need for that.

24  population of 9.1 billion (UN/ISDR, 2007; FAO, 2009

**I checked and found World Population Clock 2 May 2022: They give  7.9 Billion People (2022) – Worldometer**

I meant the projected population of 9.1 billion by 2050. The new projection is 10 billion by 2050. I fixed that in the text: "… by 2050 to feed the projected population of 10 billion"

Table 1

**What do these numbers mean?  Please make your caption more informative.  Add PET = Potential ET.  Rotate the table so we don't have to crane our necks! It fits if you make the columns a bit thinner and deeper.**

 I broke Table 1 into two tables, so both are rotated and easy to read. The first table shows the land use, which is very easy to understand. The second table is described in the text. I added Potential ET to PET in the caption of Table 2.

**At this stage, I have copied as many acronyms that I can find, some of them having their meaning explained, but after listing 23 at this stage, I plead for a list of acronyms that we can check out after the conclusion, before Refs.**

 A list of acronyms is added at the end of the manuscript.

149  ……0.1 is the scale factor meaning that the data had to be corrected by multiplying them by 0.1

(Running et al., 2019).

**I do not understand this sentence; reducing the data by a factor of 10? What for?**

This is because of the way MODIS data are gathered. It was out of my control, and only by reading the manuals, I was able to figure that out. I do think it is too much unnecessary detail, and to avoid confusion, I decided to remove this sentence.

167 V280 combined with the MSWX product (?)

$ In place of (?) I suggest "as they match in frequency (3 hr) and pixel size (0.10)"

I actually got this part clarified as "we will use MSWEP combined with MSWX to provide real-time forecasts {Beck_H.E.-2022-01}."

175

**Getting info from the farmers is very smart**

I agree.

Fig.2

**Good informative layout**

Thank you.

2.5 Relative importance of FarmCan inputs to P

**Does not make sense - inputs FROM Precipitation ?**

Thank you for pointing this issue out. I removed the P from the title, I think it was confusing.

230 variables (ET, PET, SM, and RZSM) are used first as predictants

**predictant is not a word in the Oxford English Dictionary, nor could I find it on the Web. Nice try but you might substitute:"seen as items to be estimated"**

This word is used very commonly in other articles, especially the ones in the statistics and data science domain. If you search for it, several articles and presentations come up

with this name as a synonym for "predictor". I know it might sound unfamiliar, as I too, had to learn it the first time. But I have seen it enough to now use it.

Fig. 3

**That is seriously good  corroboration cell for cell - almost identical - by eye (I did it in one minute) and I would estimate a cross correlation average of 95%**

 Thanks.

Figure 4. Spatial patterns of climatology. Data was collected from 2015-2020 for the agricultural months (Apr-Oct).

That is pretty much all we had back then for SMAP; yet, several articles refer to this as the climatology that SMAP can show so far. I have removed the word " climatology" for the consideration of the short record to avoid confusion.

**Please expand the legend in this relatively short article, as most readers will check the abstract, then possibly the figures which need to be self-explanatory.  Then they might take the challenge of the text if they have been enticed!  Expand the acronyms here, as well as listing them at the end of the text.**

 I have done that in the new manuscript and ensured all the figures are explained enough. More explanation for each figure is also brought in the text.

Fig. 5

**In the caption please change Apr—Oct to "April and October".  Also, please give horizontal definition of columns in legend - it took me a while to unpack …**

I change Apr-Oct to April to October. I reproduced all the plots and made the fonts more significant, changed the labels, and explained the plots extensively in the text.

Fig. 6

**Make these sample bars thicker as in the figure - their colours are indistinguishable in this legend;**

I have done it in the new manuscript.

Fig. 7 gets it right.  What is 'Teal'? Light green? Make the bar-chart thicker?  The dates are unpackable - they are a jumble. In my first look I had no clue as to which is day, month nor year and what the numbers below the blank spaces are designed to tell the reader. Why not give dates, of start and finish, of the readings?

 I regenerated this and made sure the dates read precisely as intended, the legends are more prominent and hopefully easier to understand.

Fig. 8

**What about (b) & (c).  Nevertheless, our figures are well laid out imbedded in the text. Also, the 3 & D are chopped off ...  the images are very readable and can be reduced in size without loss of message - same for Fig. 7 which I missed**

 Yes, I made sure nothing was chopped off.

Fig. 9

**Enlarge the words Predicted as  they are unreadable at an A4 size - Observed as well. There's enough space.  Also please make the caption more informative**

 Instead of putting the labels in the figure, I just explained them in the caption of the figure. This was because it was confusing for some reviewers that NI was also predicted, which was not. NI is calculated from predicted values.

4 Conclusions#

**Most potential readers will probably scan the Abstract, look at the Figures and possibly read the Conclusion, before they decide to read the whole.  Please repeat the text referred to by the acronyms in this passage.   Because you have a lot of them, please add them in an appendix for reference below the text.  Your article is relatively short at 350 lines including Figs & Tables, so an extra page will not hurt!**

I added a list of acronyms.

Thank you for your time and care in writing a comprehensive review.